# The Persistent Paradox of Rapid Eye Movement Sleep (REMS): Brain Waves and Dreaming

**DOI:** 10.3390/brainsci14070622

**Published:** 2024-06-21

**Authors:** J. F. Pagel

**Affiliations:** 1Family Medicine Department, University of Colorado Medical School System, P.O. Box 6, Arroyo Seco, NM 87514, USA; pueo34@juno.com; Tel.: +1-719251707; 2Psychology Department, Cape Breton University, 38 Gull Cove Rd., Glace Bay, NS B1K 3S6, Canada

**Keywords:** rapid eye movement sleep (REMS), dream, sleep, theta, electrophysiology, parasomnias, nightmares, resonance, brain waves, neuroelectrophysiology, bizarreness, gamma, dream recall, consciousness, neuroconsciousness, paralysis, lucidity, paradoxical, dream content

## Abstract

The original conceptualization of REM sleep as paradoxical sleep was based on its EEG resembling wakefulness and its association with dreaming. Over time, the concept of paradox was expanded to include various associations with REM sleep, such as dream exclusivity, high recall, and pathophysiology. However, none of these associations are unique to REM sleep; they can also occur in other sleep states. Today, after more than fifty years of focused research, two aspects of REMS clearly retain paradoxical exclusivity. Despite the persistent contention that the EEG of human REMS consists of wake-like, low-voltage, non-synchronous electrical discharges, REMS is based on and defined by the intracranial electrical presence of 5–8 Hz. theta, which has always been the marker of REMS in other animals. The wake-like EEG used to define REMS on human polysomnography is secondary to a generalized absence of electrophysiological waveforms because the strong waves of intracranial theta do not propagate to scalp electrodes placed outside the skull. It is a persistent paradox that the theta frequency is restricted to a cyclical intracranial dynamic that does not extend beyond the lining of the brain. REMS has a persistent association with narratively long and salient dream reports. However, the extension of this finding to equate REMS with dreaming led to a foundational error in neuroscientific logic. Major theories and clinical approaches were built upon this belief despite clear evidence that dreaming is reported throughout sleep in definingly different physiologic and phenomenological forms. Few studies have addressed the differences between the dreams reported from the different stages of sleep so that today, the most paradoxical aspect of REMS dreaming may be how little the state has actually been studied. An assessment of the differences in dreaming between sleep stages could provide valuable insights into how dreaming relates to the underlying brain activity and physiological processes occurring during each stage. The brain waves and dreams of REMS persist as being paradoxically unique and different from waking and the other states of sleep consciousness.

## 1. Introduction

The brain waves and dreams of REMS persist as being paradoxically unique and different from waking and the other states of sleep consciousness. Today, the most paradoxical aspect of REMS dreaming may be how little the state has actually been studied. Few studies have addressed the differences between the dreams reported from different stages of sleep. Addressing the similarities and differences between electrophysiology and dreaming of different forms of sleep-associated consciousness provides valuable insight into how rapid eye movement sleep (REMS) dreaming relates to underlying brain activity and physiological processes. 

Rapid eye movement sleep (REMS) was identified in the 1950s after Stages 1–4 had already been discovered. At that time, there was considerable debate as to what to call this physiologically unique pattern of sleep. Based on its association with both intense episodes of dreaming and activated non-synchronous EEG activity that looked much like wakefulness, the state was initially designated “paradoxical” sleep [1,2]. REMS is a state of sleep consciousness with complex anatomic, chemical, and electrophysiological physiology that is conceptually far larger than its limited attribute of rapid eye movements. This sleep/dream state became a seminal construct for the developing fields of psychoanalysis, psychology, dream science, and sleep medicine. Persistently contradictory REMS elements continue to inspire equivocation, cognitive dissonance, and research, producing logically counter-intuitive results. Today, primary aspects of this state can still be considered paradoxical. Recent research addressing its neuroelectrophysiology and associated dreaming continues to produce major changes in both applied neuroscientific theory and our comprehension of this synonymous state of paradox and REMS [3].

Early work with REMS emphasized its associated neuroanatomy. In decorticate cats, the eye movements, muscle atonia, and spatial orientation associated with REMS were maintained even when the brain was resected above the brainstem [1]. Since REMS was equated with dreaming, this brainstem-based neuroanatomic perspective was integrated with Freudian psychodynamics to produce neuroconsciousness theories of brain functioning, suggesting that dreams were emanations of the primitive Id-associated brainstem [4]. Current understandings of REMS neuroanatomy based on electrophysiologic scanning and animal models indicate that extensive areas of the cortex, hypothalamus, midbrain, and medulla are involved in the control and/or suppression of the globally projected neuroprocessing [5]. Some animal neuropsychologists have interpreted the wide spectrum of neurologic activation sites during REMS to suggest that this wake-like neuronal activity is proof that dogs dream of their waking experience [6,7]. Yet, neither REMS nor dreaming are associated with clearly described neuroanatomic markers or consistent patterns of neuronal activation [8]. Sleep and dreaming are global states defined and best described by their associated neuroelectrophysiology with neuroanatomic control and effector sites poorly described and loosely reflective of state electrophysiology [9].

## 2. REMS Paradox Part 1: The Brain Waves

Soon after its discovery, neuroscientists begin to refer to paradoxical REMS as desynchronized sleep. There were no oscillatory waveforms seen on the scalp EEG like those seen in the other stages of sleep, but rather a disordered pattern of electrical activity, a “wracked storm-tossed ocean” of intersecting spikes and poorly formed waves, much like that seen during waking. Polysomnographic monitoring using multiple sensors to detect the presence of a low-voltage, wake-like electroencephalogram (EEG), conjugate eye movements recorded by electrooculogram (EOG), and diminished chin electromyographic (EMG) activity was developed as a technique that could be used to record REM sleep in humans [9]. Today, some neuroscientists and many sleep clinicians persist in their description of human REMS electrophysiologic activity as disordered, low-voltage, and wake-like [10,11].

REMS is more evident in animal models, where it is defined by long, well-synchronized runs of recorded theta (5–8 Hz). This frequency is only minimally apparent on recordings from the scalp EEG leads used in human studies. It has even been suggested that this “paradoxical” lack of REMS synchronicity might describe a basic physiologic difference between humans and other animals [12]. 

However, the wake-like human EEG seen during REMS has turned out to describe a false dichotomy. The low-voltage asynchronous EEG associated with the human EEG is an artifact related to recording technique. Since the EEG frequencies associated with the other sleep stages easily propagate to scalp electrodes, it seemed rational to expect that the brain wave system associated with REMS, the well-described coherent frequency of theta, would also propagate beyond the brain to the scalp leads of the EEG. However, REMS theta differs from the other frequencies of sleep. It is difficult to observe and record outside the constraints of the brain container (skull, dura, and scalp). The major difference noted in the REMS EEG between animals and humans has turned out to be the electrode placement (Figure 1). Recording electrodes are implanted intracranially in animal models, and after recordings are completed, the animals are sacrificed. In human EEG recordings, the electrodes are non-invasively attached to the scalp. When electrodes were eventually placed within the human brain during intracranial surgeries for recalcitrant seizure disorders, the tightly coupled REMS theta rhythm that had been seen in other animals became readily apparent [13].

## 3. Ponto-Geniculo-Occipital (PGO) Waves

Ponto-geniculo-occipital (PGO) waves are low-voltage discrete spike/waves that can only be seen using intracranial electrodes. PGO spikes begin as electrical pulses in the pons and extend to the lateral geniculate nucleus of the thalamus, before extending to the primary visual cortex of the occipital lobe. These small signals have an amplitude of 100 microvolts (μV), last only 350 milliseconds (ms), and often occur in clusters of more than 25 spikes per minute. PGO waves often precede the onset of REM sleep, occurring 30–90 s before the first conjugate eye movements, EMG dropout, and the onset of synchronous runs of theta. This close association with REMS has led some neuroscientists to suggest that PGO spikes are the neurological trigger for REMS [14]. Based on the belief that REMS is dream sleep, it has been suggested that PGO waves are an on/off trigger for dreaming [15]. More recently, the relationship with REMS has been reconfigured so that PGO spikes are now theorized to have significant roles with hippocampal theta in visual field localization and marking [15].

## 4. QEEG and MEG

The electrical fields associated with REMS can also be recorded indirectly in humans using 256-lead quantitative QEEG systems. The QEEG signals are processed digitally, transformed, and analyzed using complex mathematical algorithms so that the otherwise hidden, low-voltage scalp signals of REMS-associated theta and gamma can be revealed [16]. QEEG recordings of the theta and gamma are obtained from short, clean windows of data that require computer-adjudicated filtering and comparative analysis to produce usable recordings [17,18]. These systems have limitations, are prone to multiple lead dysfunctions and artifacts, and produce a large data set confounded by sometimes uncontrollable variables. Gamma (35–60 Hz) recorded during REMS is also difficult to record using scalp electrodes [19]. REMS gamma is low-voltage and in the same frequency range as more powerful muscle EMG activity and the 60 Hz gamma of the environmental electrical grid. Physiologic gamma is often obscured by the 40–60 Hz block filters required for removing these artifactual frequencies from the recording. One small QEEG study is purported to describe a posterior cortical “dream center” of the brain that can be monitored to predict whether an individual will report dreams or the absence of dreaming [9]. This study is, however, minimally consistent with other work, with the proposed neural correlate of dreaming low-frequency (1–4 Hz) delta, a frequency that one of the study’s author’s had previously proposed as a marker for dreamless sleep [20]. It is also possible that low-voltage wake-like EEG activity, as in the case of lucidity, may be involved in the incorporation of sleep mentation into waking consciousness [21,22].

Attempts have also been made to use magnetoelectroencephalography (MEG) to record sleep. The earliest use of the sleep MEG was to detect gamma, a frequency theoretically tied to cellular assembly, cognitive binding, and the development of consciousness [23]. Rudolfo Llineas and Urs Ribery, among the most famous neuroscientists of this era, initially used MEG to describe 40 Hz activity during REMS. Since, at the time, REMS was viewed as dreaming, 40 Hz gamma was proposed to be a marker for dream consciousness [24]. Despite multiple attempts, other researchers have not been able to repeat these results. Neural magnetic field changes occurring in sleep are very small, and strict positional controls are required for MEG studies (few, if any, research subjects are able to attain REM sleep in the MEG laboratory) [25,26].

## 5. REMS Theta

Theta is the electrophysiologic marker for REMS. Some neuroscientists have suggested that theta does not project outside the skull because of its origin in the deep “primitive” brain structures of the hippocampus and brainstem [27]. This supposition, however, is inconsistent with the extra-cranial propagation of sigma (also a hippocampal rhythm), thetas’ lack of consistent association with QEEG vectors, and the strong theta signals observed just inside the skull on the surface of the brain. Beyond its lack of propagation, theta is also unlike the other sleep state-associated rhythms in that it has no clearly described cellular ionic origin. At the neuronal membrane, alpha is formed by the time frequency of the potassium ionic flux; sigma is tied to calcium, and delta to calcium modulated potassium channel cycling [28]. Theta seems to first appear in the rhythm modulation of hippocampal pyramidal cells [29]. It is possible that theta originates in the nano-vibrations of cellular microtubules [30]. However, it is also possible that theta has no direct intracellular origin. Global resonance theory (GRT) suggests that theta is part of a nested hierarchy of electromagnetic (EM) fields that spans the entire human physiology, encompassing both the physiologic brain rhythms and an extracerebral network of endogenous neural rhythms. These electrical frequencies can be nested in a hierarchy of resonate frequencies built upon the 5 Hz frequency of slow theta [31]. It is possible that theta forms as a resonant frequency able to entrain and form harmonic interactions with gamma and other higher-frequency fields, including ripples and gamma bursts [32,33]. 

## 6. Theta Waveforms—Operation and Functions

Animal research indicates that waking theta has concrete roles in waking cognitive functioning. In the hippocampus, theta functions as a timekeeper, a parser of memory, and an imagery marker for orientation in space. In mice and rats, theta is prominent during exploratory movements, such as when navigating through a maze looking for food [27]. During spatial navigation, the spike timing of hippocampal neurons systematically shifts to an earlier phase of the theta cycle as the rat moves across the field, a phenomenon called “phase precession” [34]. During sleep immediately after spatial tasks in which rats repeatedly run through a sequence of place fields, hippocampal neurons fire in the same sequential order as during waking [35]. Theta can set the timing of neuronal firing, playing an important role in the formation and retrieval of episodic and spatial memory [36]. Cognitive reaction time is comparatively better in animals with synchronization when compared to those with a desynchronization at the theta frequency [37]. Studies have implicated theta phase modulation in memory formation and retrieval [32]. 

## 7. The Paradox of REMS Neuroelectrophysiology

The electrophysiology of REMS continues to be a paradox but for other reasons than as initially described. The EEG of REMS has only an artifactual resemblance to the EEG of waking. REMS is defined by the presence of a synchronous electrical field: 5–8 Hz theta. The low-voltage wake-like EEG used to define REMS on human polysomnography is a null state of generalized wave absence existing secondary to the non-propagation of intracranial theta. This is the most paradoxical and unexplained persisting aspect of REMS electrophysiology. The other sleep/dream frequencies of consciousness (alpha, sigma, and delta) propagate outside the skull. REMS theta is restricted in time and space to a homunculus-like dynamic of cyclical intracranial electromagnetic fields that do not extend beyond the brain. 

## 8. REMS Paradox Part 2: Dreaming

Sleep is comprised of a group of very active central nervous system (CNS) processing states. Except for millisecond interludes of neuronal quiet during deep sleep (Stage 3), biomental activity takes place throughout sleep. This neural processing is often associated with some form of reported dreaming. 

REMS dreaming is the sleep consciousness associated with the state. For much of the last era, all dreams were presumed to have occurred during REMS. Much of the research and literature addressed dreaming as a unitary state based on the belief that REMS was the physiologic marker for dream consciousness. However, even just after the state’s discovery, it was clear that dreams were being reported from all of the stages of sleep [38]. Early studies indicated that 80% of dreaming took place in REM sleep, with 6.9% of dreams reported from the other sleep stages [39]. With improved methodology and the expansion of the definition of dreaming to include other types of mentation, dream report rates from the non-REMS stages have risen to levels greater than 70% [40]. Some individuals will report sleep-associated mentation (dreaming) whenever they are awakened [41]. REMS and dreaming are doubly dissociable [42], with REM sleep occurring without reported dreaming and dreaming reported outside REMS [43] (Table 1). The belief that REMS is dreaming is a meme and bias that taints a large portion of the research and literature addressing dreaming.

## 9. Activation-Synthesis

Activation-synthesis theory postulates that neuronal activation is the marker for brain functioning [48]. The association of intense dreaming with the “activated” neuroelectrophysiology of REMS was presented as the primary proof of activation-synthesis theory. Over the next fifty years, various versions of this theory, including AIM (activation, input–output gating, and modulation), covert dreaming, and protoconsciousness, have dominated the fields of sleep and dream neuroscience [11,48,49]. All neuroconsciousness theories have been built upon the founding construct of REMS as being equivalent to dream sleep. Despite the loss of this primary experimental proof, neuroconsciousness theories continue to delimit neuroscientific and neuroanatomic understandings of brain functioning (mind). REMS = dreaming continues to be presented as fact, incorporated into the theories of neuroconsciousness built upon activation-synthesis, basic psychology texts, neuro-medical literature addressing sleep/ dream associations, and in many of the popular sleep and dreaming texts [20,49,50,51]. REMS equals dreaming has become an apparently irrefutable meme.

## 10. Psychodynamics

At the time of the discovery of REMS, almost all of the literature addressing sleep and dreaming was based on psychoanalytic theory. REMS was construed as a “smoking gun,” proof that dreaming was a state of activation of the Freudian “Id”, a primitive neuronal system throbbing in the brainstem, activating our brains and behaviors [52]. Freud had proposed dreams as a “royal” road to the unconscious, an interpretive path that could be used to explore the psychodynamics of the human psyche [53]. Extensive literature has documented the use of dream interpretation in the diagnosis and treatment of psychiatric illness. Unfortunately, beyond anecdote, little evidence indicated that dream-based psychoanalysis actually worked to treat psychiatric illness [54]. Dream content, while reflective of an individual’s waking experience of psychiatric disturbance, was not particularly useful in diagnosis. While dream interpretation can be used to expand memory associations and explore the significance of dream stories, today, the psychoanalytic use of dream content exists primarily in approaches to self-actualization, self-understanding, creativity, and art [55]. A general perspective persists, however, that dreams are strange, bizarre experiences that function in intrapersonal psychodynamics and cognitive CNS operations.

## 11. REMS and Dreaming: The Special Relationship

Over the last seventy years, the physiology of REMS has been the focus of extensive study. There is a strong record of failed attempts to maintain the REMS–dreaming correlate, but far fewer studies comparing REMS dreaming to the dreams of other sleep stages. REMS dreams continue to be construed as “special;” the classic dreams most likely to be remembered and written down, characterized by vivid imagery, bizarre or implausible storylines, potential lucidity, increased and often negative emotionality, and high levels of motivational significance [56]. They are often described as the most bizarre, most emotional, most lucid, and most significant of dreams. More recent literature has questioned definitions, the actual sleep stage association of the reported dreams, and the bias of the subject and researcher [57]. In studies strictly controlling for transference effects and bias, it has been difficult to demonstrate significant differences in REMS dream content from reports obtained from the other stages of sleep [58]. Today, it is unclear as to whether any special relationship exists between REM sleep and dreaming. While it seems likely that most of the reports of associated dream content included in the psychoanalytic literature are from REMS, dreams and dream-associated parasomnias are reported from all sleep stages. It has been argued that the typical REM sleep dream has higher recall, greater length and narrative development, greater negativity and association with nightmares, increased emotional content, and higher levels of bizarreness and salience than the dreams that are typically reported from the other stages of sleep. There is contrary evidence, however, for all these contentions, except for report length [41].

So, what remains of the “special” relationship between REMS and dreaming? Many of the accepted foundational correlates were incorrect, insubstantial, or tenuous at best so that REMS = dreaming has become an exemplary example of a foundational reference failure. What follows is a short review of recent research and theory addressing REMS and its association with dreaming. 

## 12. REMS Dream Recall

REMS is a state of high dream recall, with a dream report on awakening frequency of 70–80%, in the same range as sleep onset—Stage 1. If recall assessment is restricted to the last REMS period of the night using a sensitive QEEG system, the retrieval rate of REMS dreaming can be increased to above 90% [59]. Both REMS and dreaming are experienced into extreme old age, continuing to be reported after major levels of CNS damage [60]. When a subject is still able to consciously communicate, the loss of dreaming is not typically associated with damage to brainstem areas controlling REMS; the loss of dreaming is most often reported after significant basilar bi-frontal CNS damage such as that created by frontal lobotomy psychosurgeries [44]. 

## 13. REMS Dream Content

REMS dreams have been described as the classic, long, “special” dreams most likely to be remembered and written down, replete with vivid imagery, a bizarre or implausible storyline, potential lucidity, increased and often negative emotion, and high levels of motivational significance [61]. Freud proposed that such dreams marked a royal road into the unconscious, an interpretive path that could be used to expand memory associations, the significance of narrative stories, and explore the psychodynamics of the human psyche [53]. This psychoanalytic perspective has contributed to the belief that dreams are evolutionarily “primitive” compared to more advanced brain functioning, a point-of-view that persists in the terminology of neuroanatomy contrasting neocortex, executive, and higher brain functions, with the conceptual perspective of dreaming as an evolutionarily inferior brain stem process [62]. 

Dream content has proven difficult to study. An amazing number of variables are known to affect reported dream content. Dream content is altered by the gender of both subject and technician; the type, site and time of reporting; dream definition, bias, and expectations of both subject and tech; and socioeconomic variables, including economic, language, and educational levels. Transference is particularly difficult to control. The strongest consistent variable known to affect dream content is the individual’s waking experience. The “continuity hypothesis” describes the only variations in content that clearly persist when methodology, bias, and transference variables are eliminated [63]. In most studies of dream content, confounding variables have been ignored, so that today it is unclear whether dream content is significantly affected by age, personality, culture, psychiatric illness, creativity, psychodynamic style, and/or trauma. Recent studies suggest that there may be few differences in dream content between REMS and the dreams collected from other stages [45]. Today, explorations of dream content are utilized primarily in approaches to self-actualization, self-understanding, creativity, and art [57]. 

## 14. Report Length

The dreams and the nightmares that occur during REMS-associated theta are longer, narratively bizarre, and include more wake-like cognitive thought than the dreams of the other sleep states [64]. Dream content has been described as the thinking of the body, an example of the birth of the literary process. Dream plots, like those of literary fiction, are a continually evolving pattern of imagery and events [65]. Dream reports rarely form full narratives but more often incomplete fragments of stories that simplify past happenings by generating tight fictional narratives around associated schemas [46]. REM sleep dreams are often subjectively perceived as story-like and autobiographically meaningful [21]. The longer REMS dream reports lend themselves to literature-based narrative analysis, an approach that has been used in the attempt to prove that REMS dreams differ from non-REMS dreams in having content that is more improbable and bizarre in the domains of dream plot, cognition, and affect. 

### 14.1. Bizarreness

Bizarreness, the quality of being strange and unusual, is a matter of contention in dream science. In 1987, researchers at Harvard designed a bizarreness scale that used the storyline narrative content of a dream to rate bizarreness. Dreams that included incongruous, uncertain, and discontinuous statements were rated as the most bizarre. Since REMS dreams are longer and have greater narrative structure, studies using this scale consistently demonstrate that REMS dreams are the most bizarre [61]. Based on this rating scale, the shorter dreams of the other stages rate as less bizarre despite apparently “bizarre” characteristics such as hallucinations, extreme emotional distress, and intense disassociation from reality. An alternative bizarreness scale taking such dream-associated behaviors into account indicates that dreams reported from sleep onset (Stage 1) and deep sleep (Stage 3) may be more bizarre than the dreams of REMS [66]. The rating of bizarreness is an excellent example of how researchers can manipulate experimental methodology. 

### 14.2. Lucidity

Lucid dreaming is the experience of achieving conscious awareness of dreaming while asleep. Most dreams have lucid characteristics in that they are described from the dreamer’s point of view with the dreamer present and, at some level, aware during the mental activity of every dream. Much of the research into this state has focused on identifying and attempting to prove that lucid dreaming occurs during REMS [67]. Yet, even in early descriptions of lucidity, it was apparent that lucid dreaming took place during sigma and alpha sleep (with 18% of lucid dreams occurring at sleep onset) [68]. In the lab, some lucid dreamers can push buttons, move, or fix their gaze while asleep and use this change in gaze to signal an external observer [69]. This volitional signaling capability has allowed researchers to analyze lucid dreaming using real-time scanning systems. The initial hope was that scanning would display specific areas of the CNS active during REMS. However, the results of such studies have been confusing. Signaling is almost always associated with arousal from sleep, in which the dreamer consciously reinstitutes the perceptual and motor controls associated with wakefulness to signal researchers. Lucidity with signaling may be best viewed as a form of conscious sleep offset—the opposite of the turning away from waking consciousness required for sleep onset [22]. During lucid dreaming, there are bursts of alpha and gamma, and activation of CNS sites involved in working memory and the analysis of visual perception—brain areas normally de-activated during REMS [70]. Lucid dreaming is a transitional state between different stages of sleep and waking, a type of dreaming that has been reported from all stages of sleep except deep sleep delta. 

## 15. REMS Physiology and Dreaming

The REMS state differs from the other forms of sleep/dream consciousnesses in its association with conjugate eye movements, the suppression of skeletal muscle activity (EMG), sporadic muscle twitches, heat loss, accelerated gastric transit, hypoventilation, autonomic nervous system activation, genital arousal, increased heart rate variability, and oscillations in body temperature [71,72]. These physiologic characteristics, however, cannot be considered REMS state-specific. REMS can occur without these attributes, which also occur, albeit less often and in associated forms, in the other stages of sleep [73] (Table 2). The dreaming-associated REMS parasomnias of sleep paralysis, nightmares, dream hallucinations, and even the acting out of dreams associated with REMS behavior disorder (RBD) have all been described in association with the other stages of sleep.

While there is good evidence for the continuity of waking experience with dream content, the relationship between sleep-associated physiologic events and dream content is less clear. The muscle twitches, disordered breathing, apneas, temperature changes, and gastric reflux associated with REMS are rarely incorporated into dream content [74]. REMS-associated genital erections and muscular paralysis, however, have an extended history and diagnostic association with both REMS dreaming and dream-associated parasomnias.

## 16. REMS Associated Genital Arousal

Psychoanalytic dream theory posits that sexual “wet” dreams occur during REMS as a reflection of brainstem activation and the influence of the primitive “Id.” Yet, beyond anecdotal reports, there is little evidence supporting the association between REMS genital erections and sexually thematic dreams [78]. For many years, sleep studies were used to prove or disprove the presumed psychogenic origin of male impotence using strain gauges that could detect penile tumescence during REMS. With time, it has become obvious that impotence is better viewed as a neurophysiologic dysfunction independent of etiology and best treated with phosphodiesterase type 5 inhibitors, such as Viagra. Today, nocturnal penile tumescence (NPT) studies are rarely performed and reserved for selected cases where the exact determination of the underlying case for impotence is deemed necessary. Sleep-associated genital arousals and wet dreams, like lucidity, signally, and dream report hot zones, might be better viewed as near-awakening transition experiences [79].

## 17. Electro-Muscular Suppression and Sleep Paralysis

An abrupt drop in mentalis EMG innervation is a primary polysomnographic criterion for human REMS [8]. The motor paralysis associated with REMS sleep paralysis has been tied to multiple well-described chemical and motor trigger sites in the pontomedullary reticular formation interconnected synaptically and utilizing the neurotransmitters glycine and Gamma aminobutyric acid (GABA) (+) and the neuromodulators norepinephrine and serotonin (−) [80,81] (Figure 2). Sleep paralysis presents as a dissociated state in which the REMS atonia of major skeletal muscle systems persists into wakefulness. The Parasomnia is well titled—the experience of awakening from sleep with an inability to move. Although sleep paralysis can be reported during Stage 1 (sleep onset alpha), it occurs most often on arousal from REM sleep [75]. Episodes of sleep paralysis often include negative and frightening content, developed in detail, and associated with distress that extends into awakening. Predisposing factors include sleep deprivation, irregular sleep–wake schedules, and PTSD [82].

During episodes of REM sleep behavior disorder (RBD), individuals sometimes physically act out their dreams, behaviors that can be dangerous to both the dreamer and their sleep-mate. As we currently understand RBD, the neural mechanisms that serve to disable motor activity during REMS dreaming do not function normally, resulting in the physical acting out of the dream experience. RBD, most common in older males, is closely associated with progressive neurologic illnesses, such as Parkinson’s disease [83]. Some studies have failed to find a clear correlation between RBD behaviors and described dream content [84].

## 18. Nightmares

The most experienced parasomnia is the nightmare, a REMS dream of utter and incomprehensible dread, beginning as a seemingly real and coherent dream sequence and becoming increasingly more disturbing as it unfolds [76]. The emotions of nightmares include anxiety and fear, as well as anger, rage, embarrassment, and disgust, with a nightmare’s story most often focusing on imminent physical danger [85]. Other negative dreams rarely have the length and terrifyingly real detail of the nightmare [77]. Nightmares are common experiences, with up to 70% of us having more than one per month [86]. Many nightmares induce awakening, sometimes as an escape from the storyline of the dream. Nightmares can be secondary to stress, anxiety, irregular sleep, and medication. Surprisingly, the medications inducing nightmares at the highest frequency increase CNS levels of serotonin, a REMS-suppressant neurochemical [47]. The most common exogenous cause of nightmares is unreconciled trauma. Nightmares are the most common symptom of post-traumatic stress disorder (PTSD), a diagnosis in which up to 50% include re-experience of the trauma. Except for the notable exception of those individuals with the diagnosis of PTSD, nightmares occur only during REMS [87]. 

## 19. SUMMARY: Brain Waves and Dreaming, The Paradoxes of REMS

The original conception of REMS as paradoxical sleep was based on EEG electrophysiology resembling the “activated” EEG of waking and the association of REMS with non-perceptual sleep-associated consciousness (dreaming). Through the years, the concept of the REMS paradox expanded to include concepts as diverse as sleep-associated dream consciousness, neuroconsciousness, parasomnias, lucidity, sleep paralysis, wet dreams, and emotional neuroprocessing. While each of these constructs has at some point been considered exclusive to REMS, in every case, evidence indicates that equivalent cognitions and behaviors take place in other states of sleep and waking consciousness (Table 2). The REMS state is associated with a loose pattern of non-exclusive correlates, so that attempts to apply scientific metaphors to REMS often produce unclear, contradictory findings and paradoxes [66]. 

### 19.1. Neuroelectrophysiology

While the association of REMS with the electrophysiology of waking consciousness is provably incorrect, philosophically, this construct persists in neuroanatomic semiotics of executive and higher brain functioning, and in the cognitive and electrophysiologic concept of activation. There is a clear bias to equate higher more-activated frequencies, such as gamma, with functions of consciousness. Consistently, the ‘on’ state of activated neurons is viewed as the operative phase of neuronal functioning. Yet, there is good evidence that off and periodically disconnected states, such as deep sleep dreaming and the default waking state of mind-wandering, are states with important roles in cognitive processing and functioning [77,88]. Despite such evidence, such off and disconnected default states have received minimal study.

CNS electrophysiology is potentially as complex and important for physiologic functioning as cerebral neuroanatomy [41,89,90]. Theta functions in the hippocampus as a time-keeping marker for orientation in space and memory [36,91]. Theta is likely to have other functions as it interacts as a base frequency in resonance with short higher-frequency ripples and bursts, as well as with the slower frequencies of respiration, circulation, sympathetic and parasympathetic discharge, and the circadian and ultra-circadian endocrine conveyed patterns of light [33,89]. REMS theta is formed in intracerebral isolation independent of neuronal ionic time sequences and protected from environmental contamination by the same system that confines it within the protective structure of the skull and the dura. An electromagnetic homunculus forms in the central brain during episodes of REMS. This fluxing area of hippocampal theta induces the neuronal net to fire in concert with the polarizing upstroke of the waveform so that the firing of each affected neuron augments the field. There are billions of neurons available, and the wave frequency builds on the summation of neural spikes. Synchronous extracellular field potentials develop, go through feedback loops, affect neurons, and propagate throughout the CNS [28]. The complex series of neuronal spikes utilized digitally in creating each wave can become information incorporated into the traveling wave, sometimes referred to as phase coding. At distant CNS sites, that pattern of neuronal firing can be reproduced, reflected, and reconstituted into a complementary series of the same neural firing pattern that was initially utilized in producing the wave [92]. Some recent studies suggest that neuroelectrophysiology is of even greater complexity, with multiple forms of sleep consciousness present in the CNS at the same time, competing for available resources and affecting the functioning of different neural network systems in the CNS [93].

### 19.2. REMS Dreaming

Equating REMS with dreaming was a foundational error of logic. Almost all studies of dream content addressed dreaming as a unified state, and major theories of neuroscience were built upon this loose and provably false correlation. Research into dreaming became focused on REMS as the physiologic marker for dreaming, and thus, the differences in dreaming between the different sleep stages were rarely addressed. The profound differences in the dreams reported from the different sleep stages are perhaps most easily described by focusing on the differences between sleep-state-associated types of frightening dreams. The visual, high recall, creatively frightening sleep onset hallucinations of sleep onset Stage 1 differ markedly from the brief moments of terror we call panic attacks that occur in light sleep (Stage 2). The intense, bizarre, confused arousals of Stage 3 deep sleep are profoundly different from the other states of sleep and waking. And these intensely frightening and sometimes extraordinarily bizarre dreams have little in common with the reality-like, narrative, and complex storylines of REMS nightmares. Few studies have compared the differences between the non-frightening dreams of the various sleep stages, but they are also associated with profoundly different EEG activity, physiology, content, and patterns of neural activation. 

REMS is a state that became defined by the belief that REMS was equivalent to dreaming, a construct adapted to support the theoretical understandings of sleep, dreams, and neuroconsciousness. As noted in this brief presentation, almost all REMS-exclusive beliefs were conflated. Lucidity, bizarre content, intense emotion, paralysis, genital erections, autonomic disarray, dispersed brain activation, and eye movements are associated with but not restricted to the state of REMS.

## 20. Conclusions: REMS the Persistent Paradox

Early sleep neuroscientists were among the first to step outside the presumed role of dreaming in psychodynamics proposed by Freud, suggesting that the electrophysiologic aspects of REMS might induce phenomenological aspects of the associated dreaming. Allen Rechtschaffen published his now-famous paper on the single-mindedness and isolation of dreams [94]. David Foulks emphasized the intrapersonal processing of associative memory systems during dreaming [39]. However, their work was deemphasized by neuroscientists who fully equated REMS with dreaming as a unitary state. Today, after more than fifty years of confusing REMS with dreaming and the therapeutic fallacies of dream interpretation, their insights can seem prescient. The two aspects of the state that have retained their exclusivity to REMS are the state’s unique electrophysiology and the long dreams with intense intrapersonal content originally associated with the term paradoxical. REMS—the paradoxical state—persists as a path that can be utilized to understand both dreaming and consciousness. 

## 21. Future Explorations

As originally conceived, the wake-like EEG of REMS was an artifact of the technique of recording. Yet, the reason for this slewed perspective, the non-propagation of the REMS theta, retains its paradoxical fascination. The initial Berger EEG recordings from the 1920s were recordings of 10 Hz. Alpha [95]. Alpha and the other physiologic brain wave systems (sigma and delta) are easily detected on electrodes placed outside the brain. These frequencies have a clear intracellular origin based on neuronal ionic transport systems. All can be affected by applied external electromagnetic fields. REMS theta differs. It is not harmonic with the other physiologic CNS electrical fields, with the rhythms varying on a logarithmic base so that they minimally affect one another [89]. Theta exists independently of the other frequencies of sleep consciousness. Yet, theta interacts within a nested spectrum of other physiologic and consciousness-associated frequencies [33,90]. The theta of REMS is clearly very different, existing in a very different electrophysiologic environment from the other frequencies used to define the conscious states of sleep.

## 22. REMS Research Limitations

Since REMS theta does not propagate outside the brain, it is difficult to observe using scalp electrodes. This limitation has restricted human REMS studies to those that define REMS based on the absence of associated EEG frequencies on scalp EEG leads. Intracranial electrodes are required to directly observe REMS theta. QEEG can be utilized to retrospectively detect REMS-associated theta and gamma that is otherwise visually hidden in the recording. These systems are complex and produce a huge amount of filtered data that are difficult to assess using current statistical techniques. MEG has been used to detect REMS theta; however, methodologic limitations make it difficult for individuals to sleep when attached to the equipment. Without ready alternatives, in the near future, low-voltage, theta-absent EEG, recorded from scalp electrodes coupled with the presence of lowered EMG and conjugate eye movements, will continue to be used to describe the presence of REMS in humans. This derivative approach can be used to identify the occurrence of REMS but provides little information about the vagaries of its important and unique system of brain waves.

Up to this point, almost all studies have addressed dreaming as a ubiquitous, undifferentiated form of consciousness; therefore, little work has considered how dream-associated neuroprocessing, physiology, and content might differ between sleep stages. It is unclear how previous work from the era of REMS = dreaming might best be applied to understanding the dream forms and thought content collected from the physiologically and phenomenologically different states of sleep.

## 23. Conclusions

Perhaps the most paradoxical aspect of REMS dreaming is how little we actually know about the state. REMS dreams are likely to be those that we remember most often, those incorporated into our literature, and those transformative dreams more likely to be analyzed and interpreted. REMS dreams are among those most likely to be recalled, to be associated with sleep paralysis, classic nightmares, lucidity, genital erections, and conjugate eye movements. The content of REMS dreams may be the most bizarre, single-minded, and the most salient, associated with long reports that potentially reflect the psychodynamics of our functioning consciousness. Yet, none of these characteristics are exclusive to REMS.

Dreaming occurs and is reported throughout sleep. Clear evidence for the uniqueness of REMS dreaming is limited to longer report length (a variable that contributes to the narrative aspects of REMS dream reports), and the association of REMS with classic nightmares (in individuals without the diagnosis of PTSD). There have been few studies addressing the non-unitary nature of dreaming—the differences between the dreams reported from the different stages of sleep. This has produced a corpus of literature in which the mental activity associated with very different states of consciousness is conflated into something loosely referred to as dreaming. Today, dreaming is used to describe the metaphors of our existence, the poetics of literature, and the essence of our hopes and beliefs. REMS science has lagged far behind, as have the theories and constructs built upon the defined attributes and content of its associated dreaming. There has been little work directly addressing the electrophysiology of human REMS or the phenomenological characteristics of REMS dreams. The dreams of the different states of sleep consciousness are profoundly different. An exploration of those differences provides a path that can be used to understand the forms of human consciousness.

## Figures and Tables

**Figure 1 brainsci-14-00622-f001:**
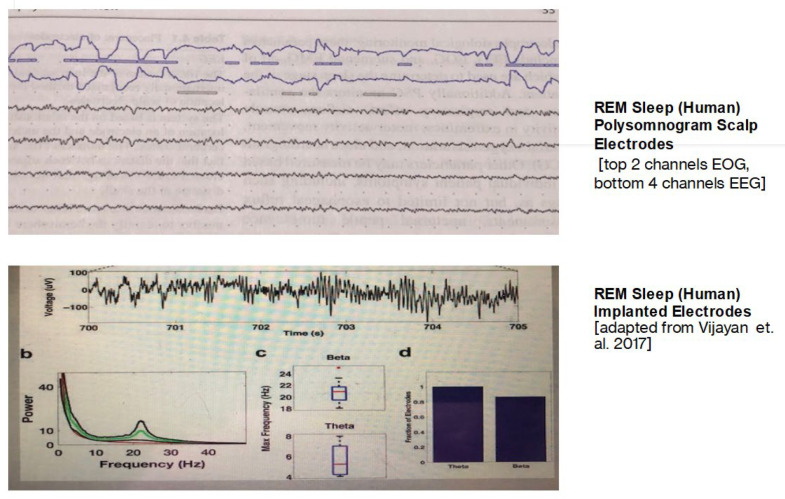
Comparative REMS monitoring. Adapted from Vijayan et al. [13].

**Figure 2 brainsci-14-00622-f002:**
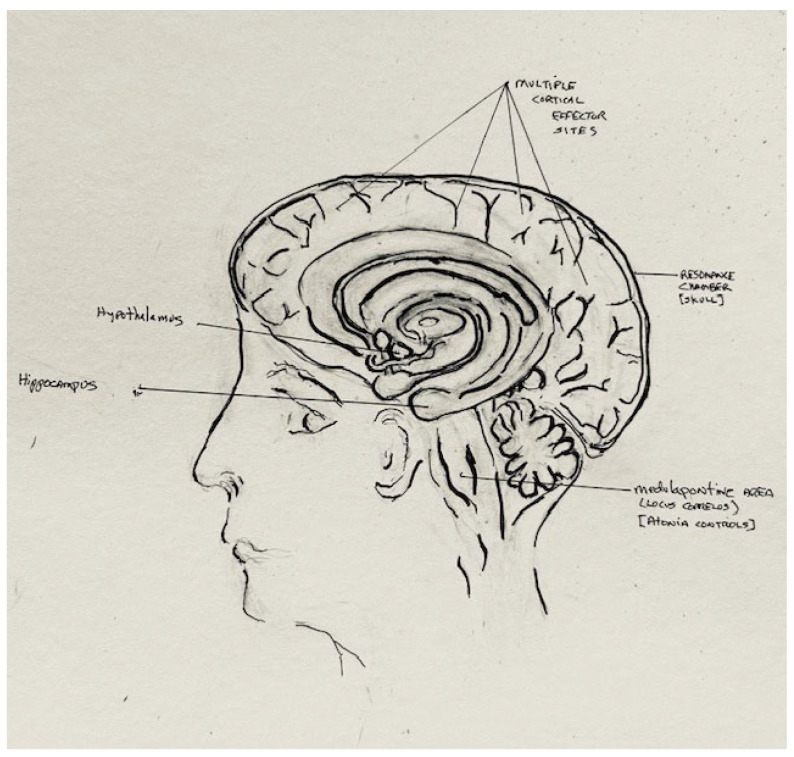
REMS dreaming: associated neuroanatomy [art work—Pagel & Broyles 2024].

**Table 1 brainsci-14-00622-t001:** Summary of primary evidence approaches that indicate that REMS is not equivalent to dreaming [8,32,40,41,43,44,45,46,47].

Dreaming occurs and is reported throughout sleep [Dreaming is reported from all sleep stages]. REMS occurs without associated dreaming. Each sleep stage is associated with state-specific forms of dreaming. Sleep state-specific parasomnias are associated with profoundly different forms of frightening dreaming. Medications and illnesses that affect REMS do not necessarily affect reported dream recall or content. Scanning systems indicate that a wide spectrum of different CNS activity is associated with dreaming during all stages of sleep. Damage to brainstem CNS areas involved in controlling REMS does not affect dreaming if the patient can report. Damage to basal-frontal areas of the brain that affect dream recall does not affect REMS.

**Table 2 brainsci-14-00622-t002:** Characteristic REMS-associated phenomenology and physiology.

	REMS Association	Other Sleep State Associations	Associated Pathophysiology
Eye movements	Defining (conjugate movements)	Rolling eye movements	Eye movement pathologies
High dream recall (>70% awakenings)	Increases through night	S1 and S3 parasomnias	
Long report length	Exclusive to REMS		
Extreme emotions	Nightmares and sleep paralysis	S3 night terrors and confusional arousals	S1 hallucinations, PTSD; S2 panic attacks
Use in creativity	Incubation	S1—induced images; S3—parasomnias	
Salience	Transcendent, persistent, and life-changing	S1 hallucinations and S3 night terrors	
Bizarreness	Narrative	S2, S3 behavioral, S1 hyynogognic hallucinations	Psychiatric hallucinations
Lucidity	Predominate state	S1, S2, and sleep offset	Confusion, psychosis
Genital arousal	Males and females	Near awakening	Drug-induced sexsomnia
Autonomic system destabilization	Predominate state	Can occur throughout sleep	Sleep position, reflux, SIDS
EMG suppression	Defining	Sleep onset S1	RBD: EMG suppression lost
Sleep paralysis	Most often	Common—S1	Narcolepsy
Dream acting-out	Most often	S3 parasonias	RBD, PTSD, Parkinson’s disease
Nightmares	Exclusive to REMS		Throughout sleep in PTSD
Cardiac irregularity	Characteristic		Associated with apneas
Hypoventilation	Increased in some	Throughout sleep	Central apneas, altitude, Cardiopulmonary disease
EEG spike waves	Suppressed		Seizure disorders
Periodic limb movements	Suppressed	Throughout sleep	Restless leg syndrome
Gastric transit	Accelerated		Reflux, irritable bowel
Body temperature changes	Characteristic	Throughout sleep	Systemic illness

Key: sleep onset Stage 1 (S1); light sleep Stage 2 (S2); deep sleep (S3); rapid eye movement sleep (REMS); REMS behavior disorder (RBD); post-traumatic stress disorder (PTSD); sudden infant death syndrome (SIDS); [9,67,68,70,72,73,74,75,76,77].

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
