# Peer review of "The Persistent Paradox of Rapid Eye Movement Sleep (REMS): Brain Waves and Dreaming"

_brainsci, 2024, doi:10.3390/brainsci14070622_

Round 1
Reviewer 1 Report
Comments and Suggestions for Authors
The original conceptualization of REM sleep as paradoxical sleep was based on its EEG resembling wakefulness and its association with dreaming. Over time, the concept of paradox expanded to include various associations with REM sleep, such as dream exclusivity, high recall, and pathophysiological aspects. However, none of these associations are exclusive to REM sleep; they can also occur in other sleep states. Manuscript has been well written, but it requires questions to be addressed.
1. Understanding the differences in dream experiences across various sleep stages could provide valuable insights into the functions and mechanisms of different stages of sleep. It could shed light on how dreaming relates to the underlying brain activity and physiological processes occurring during each stage.
2. Despite decades of research, there are still many unanswered questions about the precise mechanisms underlying REM sleep and its relationship to dreaming. Further investigation into the neural correlates of dreaming, as well as the functions of REM sleep itself, is necessary to unravel this intriguing aspect of sleep physiology and psychology.
3. Add the limitation of this study
4. “During lucid 356 dreaming, there are bursts of alpha and gamma, and activation of CNS sites involved in 357 working memory and the analysis of visual perception - brain areas normally de-activated 358 during REMS”. Add the descriptive statement with recent references.
5. What is the neural circuit mechanism for the statement “Electromuscular Suppression and Paralysis”
6. Write advantages and limitations of using QEEG and MEG.
7. Elaborate the topic “The Paradox of REMS Neuroelectrophysiology”
8. Provide a schematic figure.
Comments on the Quality of English Language
N/A
Author Response
Response to Critique: Thank you for your astute review and comments. In my opinion this paper is far better after the incorporation of your insights. Specific changes:
- Understanding the differences in dream experiences across various sleep stages could provide valuable insights into the functions and mechanisms of different stages of sleep. It could shed light on how dreaming relates to the underlying brain activity and physiological processes occurring during each stage.
incorporated
- Despite decades of research, there are still many unanswered questions about the precise mechanisms underlying REM sleep and its relationship to dreaming. Further investigation into the neural correlates of dreaming, as well as the functions of REM sleep itself, is necessary to unravel this intriguing aspect of sleep physiology and psychology.
incorporated
- Add the limitation of this study
Section added
- “During lucid 356 dreaming, there are bursts of alpha and gamma, and activation of CNS sites involved in 357 working memory and the analysis of visual perception - brain areas normally de-activated 358 during REMS”. Add the descriptive statement with recent references.
Incorporated in text and in Fig 1
- What is the neural circuit mechanism for the statement “Electromuscular Suppression and Paralysis”
Incorporated in text
- Write advantages and limitations of using QEEG and MEG.
Incorporated in limitations section
- Elaborate the topic “The Paradox of REMS Neuroelectrophysiology”
Summary is restructured to include a separate section neuroelectrophysiology
- Provide a schematic figure.
Included – Fig. 1
Again thanks wor your astute and well written comments
JF Pagel
Reviewer 2 Report
Comments and Suggestions for Authors
Dear JF Pagel,
thank you for your manuscript. Regrettably, I was unable to detect the purpose of your manusript. Is it supposed to be a summary of REMS or do you propose a more differntiated perspective of REMS?
Furthermore, the manuscript held a high number of redundancies. Some arguments were difficult to understand for readers not deeply involved in sleep research and some aspects/constructs were not explained sufficiently to understand your arguments.
The manuscript would highly profit from a stringent structure to heighten understanding.
Lastly, your reference list uses different formats for citation and the manuscript holds typing and format errors.
Best regards
Comments on the Quality of English LanguageEnglish language adequate
Author Response
Response to Critique:
thank you for your manuscript. Regrettably, I was unable to detect the purpose of your manusript. Is it supposed to be a summary of REMS or do you propose a more differntiated perspective of REMS?
Interesting. This paper addresses the brain waves and dreaming associated with the REMS state. After more than fifty years of focused research, summaries of REMS research have been attempted in several book length presentations (ex. Mallick BN. Pandi-Perumal SR. McCarley RW. Morrison AR. (Eds.) Rapid Eye Movement Sleep - Regulation and Function, Cambridge UK, Cambridge University Press). Such approaches have included minimal focus on the electrophysiology and the unique nature of REMS-associated dreaming. This short paper is an attempt to integrate new understandings of REMS electrophysiology and the differences in dream phenomenology from the different states of sleep into the greater corpus of REMS literature. Understanding the differences in dream experiences across various sleep stages could provide valuable insights into the functions and mechanisms of different stages of sleep sheding light on how dreaming relates to the underlying brain activity and physiological processes occurring during each stage. There are still many unanswered questions about the precise mechanisms underlying REM sleep and its relationship to dreaming. Further investigation into the neural correlates of dreaming, as well as the functions of REM sleep itself, is necessary to unravel this intriguing aspect of sleep physiology and psychology.
Furthermore, the manuscript held a high number of redundancies. Some arguments were difficult to understand for readers not deeply involved in sleep research and some aspects/constructs were not explained sufficiently to understand your arguments.
The paper has been extensively restructured in the attempt to reduce redundancies and increase clarity while maintaining the breath of analysis of this complex state. Please note the changes in text.
The manuscript would highly profit from a stringent structure to heighten understanding.
As above. Thanks for your assistance.
Lastly, your reference list uses different formats for citation and the manuscript holds typing and format errors.
Noted and corrected.
Reviewer 3 Report
Comments and Suggestions for Authors
Pagel's paper is a very interesting narrative review that explores numerous aspects of REM sleep, including some of its physiological peculiarities and its link to much investigated phenomena such as dreams. The work shows the Author's deep enthusiasm and knowledge on the topic and will be of interest to researchers in the field. I only have a few suggestions for the Author:
- Please always put the full name first and then the abbreviation;
- Some periods in the text (e.g. lines 121-125) have incorrect punctuation and seem interrupted/incomplete; please check that all periods are formulated correctly;
- Please quote the references in the format required by the Review; also, add the statement sections at the end and your affiliations at the beginning;
- Some sections of the text should be reorganised, trying to limit redundancies and better organising the explanations provided, so as to make the sections flow more smoothly and with less repetition;
- I suggest adding a short paragraph, perhaps with an illustrative image, explaining the anatomy of sleep (the major areas involved and their connections) and a summary outline of the NREM and REM phases and their main characteristics; I also suggest providing some notion of the nueromodulators most involved in REM sleep and dream phenomena, so as to make the neurophysiology of these mechanisms clearer;
- Line 92: I suggest replacing ‘recalcitrant’ with a more appropriate term;
- Table 1 requires the inclusion of the quotations used;
- I suggest inserting a short paragraph explaining the anatomy, neurophysiology and rhythms of the hippocampus, also explaining its fundamental connections; e.g. what could be the role of cerebellar connections? Recent work suggests a role also of the cerebellum in the contexts investigated;
- I suggest two aspects that I think might be useful and have not been explored: first, very recent works have not only explored the correlations between the Defensive Activation Theory (DAT) and Active Inference Theory (AIT) in the context of sleep, but also proposed an interpretation on dreams and evaluated the possible functional significance of certain pathological manifestations, e.g., positive neurological symptoms, RBD, in this new framework. I believe that these elements may provide new interpretative insights into the evidence discussed; moreover, in the same context, it would be useful to further examine significant alterations in REM and dreams in certain pathological contexts with a brief mention. For example, in schizophrenia dreams are altered, what could be the reasons?
- The sections summarising the characteristics of dreams (e.g. lucidity, bizarre) could benefit from a Summary Table to slim down the main text and make the evidence presented clearer.
Author Response
Reviewer response. Thank you for your commentary. Specific corrections follow:
- Please always put the full name first and then the abbreviation;
Noted and corrected
- Some periods in the text (e.g. lines 121-125) have incorrect punctuation and seem interrupted/incomplete; please check that all periods are formulated correctly;
Noted and corrected
- Please quote the references in the format required by the Review; also, add the statement sections at the end and your affiliations at the beginning;
Referencing is redone as per journal requirements
- Some sections of the text should be reorganised, trying to limit redundancies and better organising the explanations provided, so as to make the sections flow more smoothly and with less repetition;
Extensive reorganization has been done as based on reviewer suggestions. Please note in text.
- I suggest adding a short paragraph, perhaps with an illustrative image, explaining the anatomy of sleep (the major areas involved and their connections) and a summary outline of the NREM and REM phases and their main characteristics; I also suggest providing some notion of the nueromodulators most involved in REM sleep and dream phenomena, so as to make the neurophysiology of these mechanisms clearer;
The neuroanatomy and neurochemistry of REMS is at least as complex as the electrophysiology. A full book (at least) would be required to present more than a cartoon summarizing known data. Since REMS was equated with dreaming for such an extensive period, little is actually known about the neuroanatomy associated with REMS dreaming other than that it is generally reflective of the known electrophysiology as described using fMRI and MEG. The best described of associated neuroanatomy and neurochemistry (that involved in REMS atonia) has been included in text. The paradoxical finding that REMS suppressive neurochemicals induce nightmares in the clinical setting has been added to the section addressing nightmares.
- Line 92: I suggest replacing ‘recalcitrant’ with a more appropriate term;
Apologies. This term had already been removed in my rewrites but somehow remained in the text as initially submitted.
- Table 1 requires the inclusion of the quotations used;
These are not quotations but rather the author’s overview of research findings that can now be quoted. This table was designed as an overview for the non-dream scientist. It now includes references.
- I suggest inserting a short paragraph explaining the anatomy, neurophysiology and rhythms of the hippocampus, also explaining its fundamental connections; e.g. what could be the role of cerebellar connections? Recent work suggests a role also of the cerebellum in the contexts investigated;
As based on this suggestion, more neuroanatomy has been included in text. However almost all work addressing REMS has proceeded from neuroanatomy with little reference to the associated and defining electrophysiology. Since a primary purpose of this short paper is to emphasize the electrophysiologic basis of REMS, a figure emphasizing the neuroanatomy would likely be counterproductive to that purpose. An equivalently extensive review of known REMS neuroanatomy has been attempted repeatedly (ex. Mallick BN. Pandi-Perumal SR. McCarley RW. Morrison AR. (Eds.) Rapid Eye Movement Sleep - Regulation and Function, Cambridge UK, Cambridge University Press). Even at book length this approach was limited in scope. As based on your comments, however, I have indicated in text the well described neuroanatomy of REMS atonia as well as noting that electophysiologic scanning systems (fMRI and MEG) produce data used to describe the associated neuroanatomy of REMS.
- I suggest two aspects that I think might be useful and have not been explored: first, very recent works have not only explored the correlations between the Defensive Activation Theory (DAT) and Active Inference Theory (AIT) in the context of sleep, but also proposed an interpretation on dreams and evaluated the possible functional significance of certain pathological manifestations, e.g., positive neurological symptoms, RBD, in this new framework. I believe that these elements may provide new interpretative insights into the evidence discussed; moreover, in the same context, it would be useful to further examine significant alterations in REM and dreams in certain pathological contexts with a brief mention. For example, in schizophrenia dreams are altered, what could be the reasons?
This paper limits itself to clarifying descriptions of REMS electrophysiology and dreaming. The literature on REMS theory and function, like that on REMS neuroanatomy is huge and diverse. As noted in text, these approaches have sometimes been misapplied and misdirected. DAT and AIT are interpretative approaches likely to provide new insights and as such any presentation of these topics should include aspects of the paradoxical electrophysiology dream associations presented in this short paper. Hopefully that might happen, in part, as based on the publication of this paper.
Dream content is a difficult area bedeviled by extensive transference and bias variables. The most consistent factor now known to affect studies addressing content is continuity with waking experience, a variable that is very difficult to control and one that has not been incorporated well into the pathologic studies of dream content such as the schizophrenia studies that you allude to. The dream reports from schizophrenia patients reflect their waking experience. It remains somewhat unclear as to whether this is based on any change in their sleep-associated psychodynamics.
- The sections summarising the characteristics of dreams (e.g. lucidity, bizarre) could benefit from a Summary Table to slim down the main text and make the evidence presented clearer.
A summary table is added. An attempt has been made in an extensive rewrite to reduce redundancy. See rewrite of text.
Thanks for your review. Pagel
Round 2
Reviewer 2 Report
Comments and Suggestions for Authors
Dear J F Pagel,
thank you for the revised manuscript. I think it is much more structured now. Nevertheless, there are still some spelling mistakes and the references list uses different citation styles.
I also think it would be beneficial to state the purpose of your summary in the introduction.
Best regards.
Author Response
added:
The brain waves and dreams of REMS persist as being paradoxically unique and different from waking and of the other states of sleep consciousness. Today, the most paradoxical aspect of REMS dreaming may be how little the state has actually been studied. Few studies have addressed the differences between the dreams reported from different stages of sleep. Addressing similarities and differences between the electrophysiology and dreaming of the different forms of sleep associated consciousness provides valuable insight into how rapid eye movement sleep (REMS) dreaming relates to the underlying brain activity and physiological processes.
you are correct, this is helpful
A sincere effort has been made to establish a full consistency between references.
Pagel
Reviewer 3 Report
Comments and Suggestions for Authors
I thank the Reviewer for his extensive review, which has greatly improved his paper. However, I still trust that some aspects could be useful additions to help the reader understand the interesting explanations in the review and to give an interpretative context to the various sections. My suggestions are not meant to cover all the literature on the subject (I agree with the Author that this context is impossible), but I do think that some hints would increase the reader's interest in the mechanisms described. Therefore, I propose again:
- I suggest adding a short paragraph, perhaps with an illustrative image, explaining the anatomy of sleep (the major areas involved and their connections) and a summary outline of the NREM and REM phases and their main characteristics; I also suggest providing some notion of the nueromodulators most involved in REM sleep and dream phenomena, so as to make the neurophysiology of these mechanisms clearer;
- I suggest two aspects that I think might be useful and have not been explored: first, very recent works have not only explored the correlations between the Defensive Activation Theory (DAT) and Active Inference Theory (AIT) in the context of sleep, but also proposed an interpretation on dreams and evaluated the possible functional significance of certain pathological manifestations, e.g., positive neurological symptoms, RBD, in this new framework. I believe that these elements may provide new interpretative insights into the evidence discussed; moreover, in the same context, it would be useful to further examine significant alterations in REM and dreams in certain pathological contexts with a brief mention. For example, in schizophrenia dreams are altered, what could be the reasons?
I would like to point out, with regard to the second point, that these are just hints that, in my opinion, might be useful and interesting to provide an interpretative framework, but the Author may feel free to use others if in his opinion they are more up-to-date and better suited to the topic. I hope I have contributed to improving the quality of this interesting work that will certainly be of great interest to interested readers.
Author Response
- I suggest adding a short paragraph, perhaps with an illustrative image, explaining the anatomy of sleep (the major areas involved and their connections) and a summary outline of the NREM and REM phases and their main characteristics; I also suggest providing some notion of the nueromodulators most involved in REM sleep and dream phenomena, so as to make the neurophysiology of these mechanisms clearer;
A cartoon of the associated neuroanatomy is included as a figure.
Unfortunately, the neurochemistry of sleep (my area of primary expertise) is often presented simplistically. More than 60 neuro-active agents, neuromodulators and neurotransmitters are known to affect sleep (1,2). Many of these agents affect REMS. The primary described effect of psychoactive agents on dreaming is on nightmares (27 different classes of medications are known to induce nightmares) (3). For some classes the effect on REMS is opposite the effects induced on reported dreaming. My greatest concern is to not simplify this complex area. Some theorists (Walker) as based on their understanding that REMS is important for the emotional processing of trauma have authoritatively recommended that REMS suppressive agents not be used in individuals with the diagnosis of PTSD. Some of those agents (ex. antidepressants) are the only medications known to improve outcomes in this populations. Other individuals (LaBerge) based on their understanding that exogenous acetylcholine should increase REMS and since their understanding is that REMS=dreaming, suggest that Ach affecting agents with known toxicity be used to increase the potential for lucidity. There is no evidence except for anecdotal reports supporting either of these approaches (4).
- Pagel JF (2007) Medication Effects on Sleep, in Sleep and Psychosomatic Medicine, (eds.) S. R. Pandi-Perumal, Rocco R. Ruoti & M. Kramer, Informa UK, Ltd. Andover, Hampshire, pp. 109-124.
- Pagel JF (2006) Medications that Induce Sleepiness, in Sleep A Comprehensive Handbook, Ed. T Lee-Chiong, J. Wiley and Sons, Hoboken, N. J. 2006, pp. 175-182.
- Pagel JF. The Neuropharmacology of Nightmares in “Sleep and Sleep Disorders: Neuropsychopharmacologic Approach,” (Eds.) Pandi-Perumal SR. Cardinali DP. Lander M. Landes Bioscience. Landes Bioscience. Georgetown Texas, pp. 241-250.
- Pagel J. F. (2020) Post-Traumatic Stress Disorder - A Guide for Primary Care Clinicians and Therapists Springer Nature, Switzerland. ISBN 978-3-030-55908-3.
I had hoped to limit the focus of this paper to the relatively unexplored area of REMS electrophysiology rather than adding on the extensively researched and complex areas of REMS neuroanatomy and neurochemistry.
- I suggest two aspects that I think might be useful and have not been explored: first, very recent works have not only explored the correlations between the Defensive Activation Theory (DAT) and Active Inference Theory (AIT) in the context of sleep, but also proposed an interpretation on dreams and evaluated the possible functional significance of certain pathological manifestations, e.g., positive neurological symptoms, RBD, in this new framework. I believe that these elements may provide new interpretative insights into the evidence discussed; moreover, in the same context, it would be useful to further examine significant alterations in REM and dreams in certain pathological contexts with a brief mention. For example, in schizophrenia dreams are altered, what could be the reasons?
The following, as based on your recommendation, is helpful to this paper and added in text and reference
Some recent studies suggest that the neuroelectrophysiology is of even greater complexity, with multiple forms of sleep consciousness present in the CNS at the same time competing for available resources and affecting the functioning of different neural network systems in the CNS (93).
Antonioni A. Raho EM. Sensi M. Di Lorenzo F. Fadiga L. Koch G. (2024) A new perspective on positive symptoms: expression of damage or self-defence mechanism of the brain? Neurol Sci. 4:2347-2351. doi: 10.1007/s10072-024-07395-x.15:638474.
Pagel